# Design, Fabrication and Characterisation of Multi-Parameter Optical Sensors Dedicated to E-Skin Applications

**DOI:** 10.3390/s23010114

**Published:** 2022-12-23

**Authors:** Lionel Fliegans, Joseph Troughton, Valentin Divay, Sylvain Blayac, Marc Ramuz

**Affiliations:** Mines Saint-Etienne, Center CMP, Department FEL, F-13541 Gardanne, France

**Keywords:** stretchable waveguide, grating coupling, artificial skin

## Abstract

For many years there has been a strong research interest in soft electronics for artificial skin applications. However, one challenge with stretchable devices is the limited availability of high performance, stretchable, electrical conductors and semiconductors that remain stable under strain. Examples of such electronic skin require excessive amounts of wires to address each sensing element—compression force and strain—in a conventional matrix structure. Here, we present a new process for fabricating artificial skin consisting of an optical waveguide architecture, enabling wide ranging sensitivity to external mechanical compression and strain. The manufacturing process allows design of a fully stretchable polydimethylsiloxane elastomer waveguide with embedded gratings, replicated from low cost DVD-Rs. This optical artificial skin allows the detection of compression forces from 0 to 3.8 N with controllable sensitivity. It also permits monitoring of elongation deformations up to 135%. This type of stretchable optical sensor is highly robust, transparent, and presents a large sensing area while limiting the amount of wires connecting to the sensor. Thus, this optical artificial skin presents far superior mechanical properties compared to current electronic skin.

## 1. Introduction

Modern electronics are made up of diverse components and devices, the majority of which rely on semiconducting and metallic elements, making them stiff and inflexible. Electronics objects are increasingly based upon flexible electronics, a field which is now expanding to also include stretchable electronics.

Stretchable electronic systems offer important and emerging uses, including solar-cell devices, plastronics, and so-called “epidermal” electronics [1]. This last example—often referred to as electronic skin, or e-skin—is based on electronic systems capable of mechanically imperceptible integration onto human skin, and even surgical and diagnostic implementations that interact within the human body to provide advanced therapeutic capabilities [2]. Temperature, touch, and motion detection are among the sensing capabilities that have attracted significant recent interest in stretchable sensors for artificial skin applications [3]. The skin is a metabolically active organ that conducts complex physiological tasks. With a surface area of 1.7 m${}^{2}$, skin is a large organ, accounting for 15 percent of total weight [4]. The main functions of the skin include physical/chemical barrier protection, temperature sensing, and sensitivy to a large variety of mechanical deformation such as compression force, indentation, shear force, and elongation. When it comes to imitating skin, the majority of current stretchable devices are electrical, which limits their stretchability. Examples of skin-like devices have been created in recent years by focusing on pressure sensor technology on large-area, stretchable substrates [5,6]. These are mostly based on transistors with a sensitised rubber as part of the transistor gate that translate changes in applied pressure into variations in the transistor output. Microstructured, rubber-based capacitive pressure sensors, with a measured sensitivity from 3 to 71 kPa${}^{-1}$, were reported by Li et al. [7]. Due to the required metallic interconnections, these capacitive-based pressure sensors are neither conformable nor stretchable, despite their high sensitivity.

Optical sensors are extremely sensitive and can be stretchable, flexible, and conformable, making them ideal for interacting with, or replacing, human skin. Previously, a non-stretchable optical pressure sensor was demonstrated, with promising results: a pressure range of a few Pa to 100 kPa with a resolution of 10 Pa was shown [8,9]. There are also some examples of lab-on-a-chip (LOC) devices made of polydimethylsiloxane (PDMS) with integrated optical and fluidic elements [10]. These so-called opto-fluidic devices are limited to biological sensing and the stretchable qualities have rarely been considered. Optical devices capable of detecting shear and stretching stress based on the colour change, or colourimetry, have also been demonstrated [11].

Sensory transduction in the skin requires a wide sensitivity range to allow differentiation between multiple inputs, such as the roughness of a surface, pressure, and localised pain [12]. Additionally, the skin must conform and deform perfectly with the body. Human skin is capable of elongation up to 80% with a breaking stress of 15 MPa [13]. In order to mimic these mechanical properties in e-skins, silicone or polyurethane elastomers, and hydrogels are commonly used [14,15,16,17].

Fabrication of stretchable sensors based on these elastomers are technically realised in two ways. A first approach, often referred to as “island architecture”, uses regions of locally stiffened material embedded within a stretchable substrate. These islands, on the order of hundreds of microns to a few millimetres in size, can either be achieved by stiffening the elastomer itself, or by embedding another stiff material in or on the elastomer. Rigid organic or inorganic optoelectronics are then integrated on these “islands”, somewhat isolating them from the global stretching of the substrate [18,19,20,21,22]. Alternatively, fully stretchable components can be fabricated by mixing an elastomer with conductive and even semiconducting materials [23,24]. The first approach allows the use of technologically mature components in contact with an elastomer. However, the mechanical mismatch, by a factor of $10^{3}$–$10^{4}$, of the elastic moduli between these materials creates point defects under mechanical deformation that have to be properly controlled [25,26]. The lower mechanical performance is compensated for by very high electrical performance. In the second approach, the opposite is true; excellent mechanical performance is achieved, but at the cost of reduced electrical performance.

A further alternative is offered by some elastomers which can be used directly as a transducer of mechanical deformation through an optical signal [27]. PDMS elastomer is often used in this context because it has low optical attenuation, below 5 dB/cm, and is transparency over a wide wavelength range, between 300–1600 nm [28], while having suitable mechanical properties for artificial skin applications. These waveguides have different form factors depending on their purpose. From the ridge type, where the light is confined in two directions through channels of some tens of microns thicknesses [29], to planar guides of several centimetres where the light is confined in one direction of propagation. The integration of the light source, as well as the acquisition of the signal, can be achieved by direct contact with the edge of the waveguide [30,31] or at close proximity without contact by means of optical diffraction gratings [32]. The diffraction gratings are designed with a periodic structure in nanometric range. These nano-structures are commonly obtained using by soft-lithography from master moulds created by electron- or focused ion beam lithography [33,34,35], or by wrinkling techniques [36]. The use of blank DVD-Rs as the master mould for the creation of diffraction gratings has been demonstrated for biological applications [37,38] and optoelectronics devices [39], chosen as this medium is cheap, mechanically flexible, and can be handled easily. However, to date this approach has not been applied to the fabrication of stretchable optical waveguides.

Here, we present a planar PDMS waveguide fabrication technique developed using a simple, low-cost, and reproducible method for a wide range of sensors for artificial skin applications. This approach combines the advantage of highly efficient optoelectronic devices, with the mechanical properties of a stretchable elastomer. By placing the rigid components at the extremities of the artificial skin, the high performance advantages of island architectures are maintained, while producing a fully stretchable sensing area. This fabrication technique allows for simple integration, localisation, and adjustment of the diffraction gratings. These gratings are demonstrated to be efficient for coupling light into and out of the waveguide. Devices utilising such gratings for both in and out coupling areas are fabricated, and their sensitivity to mechanical deformation is explored. Further, a colourimetric sensor is demonstrated, utilising the spatial variation of embedded nano-structures under mechanical stretching to transduce strain into a colour change.

## 2. Waveguide Operation

Our artificial skin devices based on optical waveguides have a fairly simple structure. In the simplest form they comprise an elastomeric waveguide with integrated light source and photodetector. These optoelectronic components can be either integrated into the elastomer in an edge coupling scheme, or can use diffraction gratings to couple light in and out from the top surface (Figure 1). While edge coupling can produce high efficiencies, it presents significant challenges in fabrication owing to alignment requirements of the optoelectronic components with the waveguiding core. The use of a diffraction grating removes these requirements, making device fabrication far easier and more akin to traditional electronics manufacturing.

Waveguides consist of a core, which confines light, and a cladding surrounding the core. To confine the light, the refractive index of the waveguide core, $n_{wg}$, must be higher than that of the cladding, $n_{c}$. In this way, light travelling through the waveguide is reflected at the interface between the core and the cladding provided $n_{wg}\sin\left(\theta \right)\geq n_{c}$, where θ is the angle of the light from the normal, and $\theta_{c}=\arcsin\left(\frac{n_{c}}{n_{wg}}\right)$ is known as the critical angle for total internal reflection. For edge coupled light, θ can be related to the angle of the light incident on the waveguide, α, as $\sin\alpha =n_{wg}\cos\theta \geq \sqrt{n_{wg}^{2}-n_{c}^{2}}$. This sets a limit on the admittance angle at $\alpha \;=\;\arcsin\left(\sqrt{n_{wg}^{2}-n_{c}^{2}}\right)$, known as the numerical aperture, NA.

Beyond the limitation of the critical angle, the propagation of light is also defined by the propagation constant $\beta \;=\;kn_{wg}\sin\theta$, where *k* is the wavenumber (k=2π/λ). β defines the phase evolution of the light through the waveguide as it is reflected from each interface, which ultimately leads to only quantised values of β being supported by the waveguide [40]. These eigenvalues of β are known as bound modes, *m*, and can be thought of as existing as standing waves within the waveguide, having a wavelength of $\lambda_{P}=\lambda /\left(n_{wg}\sin\theta \right)=2\pi /\beta_{m}$. The value of these modes is dependent on the wavelength of the light, λ, the refractive index of the waveguide and the cladding, and the thickness of the waveguide, *D*. From the derivation of modal behaviour [40], another term can be extracted, known as the *normalised frequency*, or V number, and is given as
(1)$$V\;=\;\frac{2\pi }{\lambda }D\sqrt{n_{wg}^{2}-n_{c}^{2}}=\frac{2\pi }{\lambda }D\times \mathrm{NA}$$

The V number has significance as it defines the number of modes that the waveguide can support ($V^{2}/2$) for the so-called step-indexed waveguides used here. It is also a straightforward metric to compare the behaviour of similar waveguides with differing geometries or with variation in λ.

The discussion above has considered only the behaviour of light within the waveguide, assuming the light is perfectly in-coupled co-linearly with the guiding core. However, as mentioned, such edge coupling requires a high degree of precision in terms of aligning the optoelectronic components with the guiding core. The alternative is to couple the light to the waveguide in a so-called top-coupling regime, using either a prism or a grating to diffract the light into the core, significantly easing the fabrication challenges. Grating coupling is the more wide spread of these approaches, using a periodic pattern in the waveguide to diffract the light from near vertical to be able to propagate along the waveguide. Light incident on the grating follows the well known grating equation
(2)$$n_{wg}\sin\left(\theta \right)=n_{c}\sin\left(\alpha \right)+m\frac{\lambda }{\Lambda }$$
where Λ is the grating period, and *m* is the diffraction order, Figure 1. Since light can only propagate in the waveguide as long as $\theta \geq \theta_{c}$, a condition can be put on the incidence angle α
(3)$$\alpha \geq \arcsin\left(1-\frac{m\lambda }{n_{c}\Lambda }\right)$$

From Equation (Equation 3) it can be seen that top coupling of the 0th diffraction mode is not possible from a grating coupler as it would require $\alpha =90^{{}^{\circ}}$, however higher values of *m* can be in-coupled easily.

Given the propagation of the light is now not co-linear with the waveguide, but travels at the angle θ from Equation (2), the light can be described as having a period, P, within the waveguide, where P is calculated as P = 2Dtan(θ).

Alongside the coupling and propagation of light in the waveguide, the other important aspect to consider is the loss of power seen between the source and the detector. In general terms this can be described as the total power loss, or total attenuation, $A_{tot}$,
(4)$$A_{tot}=10\log_{10}\frac{P_{out}}{P_{in}}$$
where $P_{out}$ and $P_{in}$ are the measured power at the input and the output of the waveguide.

There are many causes of power losses in the waveguide, and these can be split into coupling losses, experienced at each end of the waveguide, and linear losses experienced throughout the guiding channel. Coupling losses include the loss of the 0th order diffraction peak, as well as scattering due to imperfections in the grating. Linear losses include scattering at imperfections in the guiding core and its interfaces with the cladding, absorption by the core material, losses from the evanescent field of the guided light, and geometrical changes that cause premature out-coupling of some of the light. It is these last features that make these waveguides suitable for use as sensors in this work.

The waveguides described here, with a thickness *D* on the order of hundreds of microns, and widths of 5 mm, are referred to as slab waveguides as they optically confine the light in just one dimension. In this case the core consists of PDMS, with refractive index $n_{wg}\;=\;1.41$ [41], and the cladding is air, $n_{c}\;=\;1$. The softness of PDMS, with Young’s modulus around 1–3 MPa [42], makes it a good choice for optical electronic skin sensors, both because it matches that of real skin well, and because this allows geometric changes under mechanical stimulation which can be detected in a waveguide.

## 3. Materials and Methods

Traditionally, fabrication of a grating structure in polymer waveguides is achieved using a master silicon mould, created by e-beam lithography; a costly and time-consuming element to make [43]. It is also possible to use small, commercially available, nanostructured silicon moulds to perform imprint lithography on the uncured elastomer [44]. Unfortunately, this process leads to inhomogeneities and unevenness within the waveguide that deteriorate the optical guiding properties and reduce the fabrication reproducibility.

Our approach to creating multimodal planar waveguides for artificial skin applications takes a much simpler route, as depicted in Figure 2A. This process utilises the large area (44 cm${}^{2}$) uniform optical diffraction grating of a blank DVD-R. The DVD-R diffraction grating is used to define the in- and out-coupling gratings of the waveguide, while the guiding core is kept smooth.

DVD-Rs are manufactured from two 580 μm thick injection moulded polycarbonate (PC) sheets sandwiching an organic dye layer (AZO, Verbatim) and a thin 200 nm layer of silver (Appendix A). The silver layer is deposited by sputtering onto the first PC layer, which is embossed with a spiral groove with a pitch of 740 nm [45]. This spiral groove acts as the diffraction grating. The rigidity and thermal resistance of PC makes it suitable a substrate for the photolithography process.

In order to create a master mould for our waveguides, two structure types are required: zones with and zones without the grating. To achieve this, first the top PC layer is removed by mechanical separation, and the dye layer dissolved by soaking in ethanol, leaving the embossed bottom PC and silver layers intact. It is important that the metallic layer maintains its physio-chemical integrity during this separation as it acts to shield the bottom PC layer from direct contact with the propylene glycol methylether acetate (PGMEA) used as photoresist solvent. After separation, a layer of photoresist is spin coated on the DVD-R, with a surface roughness of 6.5 Å, and patterned by photolithography technique to define the waveguide core, leaving the grating coupling zones clear (Figure 2C (1)). Next PDMS is spin coated on the mould, conforming to the grating where the photoresist has been removed, and remaining smooth where it remains (Figure 2C (2,3)). One factor that may affect the coupling efficiency of the gratings fabricated in this fashion is the orientation of the DVD-R spiral relative to the waveguide propagation direction. As the gratings are defined with photolithography, the alignment accuracy is dependent on the mask aligner used, and is around 1 μm in this work (MJB4 mask aligner).

Finally, the whole PDMS layer is removed from the DVD-R and precisely cut to the desired final form-factor using a CO${}_{2}$ laser (Figure 2B). Full details of this process can be found in Appendix B.

This technique allows for fabrication of highly planar devices, with the offset between the grating and the guiding core defined by the thickness of the photoresist. The nanometer offset values achievable provide excellent light propagation from the grating into the waveguide. The high quality grating zone characteristics, along with the smoothness of core zones, can be seen in Appendix A.

This simple, low cost, master mould production process is both easily reproducible, and allows rapid design changes for grating integration. Fabrication of multiple stretchable optical waveguides can be realised from a single mould as many PDMS replicas are possible. The aim of this work is to use this production process to create devices with controlled geometric parameters, and to characterise their optical transduction under different types of deformation (compression force, elongation, bending).

## 4. Results

### 4.1. Waveguide Characterisation

Before considering the sensing capabilities of the waveguides, characterisation of the waveguides themselves is necessary, in particular the power losses. For this, a laser is used as the light source at the in-coupling grating, held at an angle of $\alpha \approx 30^{{}^{\circ}}$, and a photodetector placed over the out-coupling grating. $\alpha \approx 30^{{}^{\circ}}$ is a suitable angle as it satisfies Equation (Equation 3) for the grating period of the DVD-R, Λ=740 nm and m=1. The total power loss, $A_{tot}$, is calculated by measuring the laser power immediately before the grating, and at the out-coupling grating, using the same photodetector. In this way, $P_{in}$ and $P_{out}$ of Equation (Equation 4) can be replaced with the measured current of the photodetector as $A_{tot}=10\log_{10}\left(I_{out}/I_{in}\right)$.

By varying the waveguide length, *L*, the coupling and linear losses can be extracted as the intercept and slope of a linear fit to the total loss, $A_{tot}$, with length. As seen in Figure 3A, for a 400 μm thick waveguide with an output coupling grating area of 10 × 3 mm, a coupling loss of 17 dB is measured, with a linear loss of 3.2 dB/cm. This is consistent with other work on PDMS waveguides, which show linear losses of 3.1 dB/cm at 532 nm [46]. Details of these and future measurement procedures can be found in Appendix C below.

We also considered the effect of the waveguide and grating thicknesses, i.e., respectively *D* and *d* (see Figure 1). The thickness of the grating zone, *d*, is determined by the thickness of photoresist, here targeted to be 10 μm [9.4 ± 1.0 μm; N = 50], 1 μm [0.96 ± 0.18 μm; N = 50], and 200 nm [200 ± 50 nm; N = 50]. It should be noted that *d* is the thickness of the grating zone, not the depth of the grating which is set by the shape of the DVD-R at around 50 nm (see Appendix A). As seen in Figure 3B, there is no discernible trend in the waveguide losses with *d*. This is somewhat surprising as minimising *d* may be expected to reduce coupling losses between the grating and the guiding core. This could be explained by the fact that *D* is at least 10 times thicker than *d* and thus has a significantly larger impact on waveguide losses, masking any variation caused by the value of *d*. In addition, the angle of the 1st order diffraction light, θ, is expected to be around 60${}^{{}^{\circ}}$, and so geometrically would not be expected to hit the edge of the coupling zone. Conversely, reduction in the core thickness, *D*, from 400 μm [400 ± 80 μm; N = 23] to 180 μm [180 ± 30 μm; N = 16] to 120 μm [120 ± 10 μm; N = 32], leads to markedly reduced losses in light for the same length of waveguide and grating, Figure 3B. See Appendix A for the measured thickness distributions. This reduction in losses can be attributed to greater out-coupling efficiency in thinner devices stemming from the reduced reflection period, P, of the light with smaller values of D. This means more nodes are incident on the out-coupling grating, increasing the efficiency, and decreasing the total losses.

Beyond being a good material for optical waveguides, a major advantage of PDMS is its ability to be stretched up to 140% and as well be compressed [47]. Stretchability largely fulfils the requirements for artificial skin, while compressibility is crucial when considering PDMS waveguides as compression force sensors.

### 4.2. Compression Force Sensor Characterisation

As shown elsewhere, soft PDMS waveguides can be used as compression force sensors for e-skin applications [48].

When pressure is applied to a sensor, the evanescent field of the guided light is modified due to a change in the effective cladding refractive index, $n_{eff}$. Alongside this, there may be changes in internal reflections due to critical angle variation caused by the variation in $n_{eff}$ as well as the physical deformation of the interface, and changes in the effective thickness of the waveguide due to compression.

For characterisation of the compression force sensing, a motorised force measurement unit was used to apply a force, normal to the surface. The compression force measure unit was fitted with a stylus with 2 × 5 mm flat tip (see Appendix A), and the stylus driven into the sensor at 5 mm/min from an initial non-contact position up to a maximum contact compression force of 3.8 N. In the original configuration, the waveguide used to sense the compression force does not have a solid cladding as it is surrounded by air. Thus, an object in the vicinity of the waveguide sensor could significantly modify the cladding refractive index, $n_{c}$, and generate a significant change of the guiding conditions. This resulted in a rapid response in the sensor as seen in Appendix A, but this is in response to the mere presence of the stylus, meaning the device acts more as a presence sensor than a force one. A polyethylene (PE) foam layer is therefore introduced as an interface between the stylus and the waveguide. As this foam partially acts as a local cladding, it is can be considered an integral part of the compression force sensor. It serves three distinct purposes: it addresses the proximity detection issue mentioned above, it decreases the compression rate from the test bench thanks to its compressibility, and it maintains an air layer on top of the waveguide, (Appendix A) [49]. The surface topology of the foam (with an average roughness $r_{a}$ = 45.25 μm and maximum profile height $r_{z}$ = 189 μm) also means that the interface between the PDMS core and the PE is highly discontinuous and mostly made out of air. As such, the refractive index of the cladding cannot simply be thought of as being that of the PE ($n_{PE}\;=\;1.49$), but is some combination of the PE and air. We define it as the proportion of the core surface in direct contact with the PE, so $n_{c}$ becomes $n_{eff}=x\;n_{PE}+\left(1-x\right)n_{air}$ where *x* is the proportion of the surface in contact. These measurements are described in Figure 4A.

By increasing the applied normal force on the PE foam, the proportion, *x*, of the waveguide surface in contact with the foam increases, changing the evanescent field out-coupling. At the same time, the normal force compresses the waveguide core, reducing *D*, (Appendix A). While the deconvolution of these effects is challenging, their combined effect can be seen in Figure 4A, where the normalised waveguide loss with increasing compression force is seen for two waveguide thicknesses (180μm and 400μm) at two difference incident wavelengths (450 nm and 532 nm). It can be seen that, in all cases, the loss in the waveguide increases with increasing compression force, and this is fitted with an logarithmic function of the form $A=a\log_{b}\left(x+c\right)$. Fitting results can be found in Appendix A. From these it can be seen that the thinner device, 180 μm (curves yellow and cyan), is more sensitive to small compression forces than the 400 μm device (curves magenta and green). In addition, the wavelength has a significantly greater impact on the sensitivity of the thinner device. Both of these phenomena are due to the proportional change in device thickness caused by the stylus during testing. The depth of the dent in the sensor interface is likely to be approximately the same for both devices because of the small compression forces involved, but this represents a significantly larger proportion of the total waveguide thickness for the thinner device, making it more sensitive to smaller compression forces. These differences can be captured by the normalised frequency of the waveguide, *V*, given by Equation (Equation 1). It is seen in Figure 4A that the higher *V* values give lower sensitivity to the applied normal force, well explained in other work [50]. Under five compression force cycles, with a cycle time of 60 s and a 20 s pause between cycles, a hysteresis of ≈20% of the full sensing range for the 180 μm device with input wavelength of 532 nm is observed, Figure 4B,C, which corresponded to an integrated current of 37 nA·N for a single strain cycle. The origin of this hysteresis is somewhat challenging to ascertain. It is expected that both the relaxation of the PDMS core, and the interaction of the PE foam with the PDMS at their interface, contribute. However, the relative contribution of these two factors remains unclear.

It should be noted that, for real-world application, the use of such a foam layer may not be sufficient to remove object proximity effects. If this is the case, the area outside of the foam-covered sensing zone could be covered with a cladding layer to minimise such effects [51,52].

### 4.3. Strain Sensor Characterisation

Despite being stretchable, thanks to the pure elastomer composition in the active area, these sensors still show a response to longitudinal strain that can be characterised. For characterising the strain response a 180 μm thick, a dog bone-shaped sensor is used. The dog bone shape, seen in Appendix A, is chosen to minimise changes in the coupling loss due to deformations in the diffraction gratings. The dog bone is clamped between the gratings and the guiding core to ensure the gratings experience minimal strain. The waveguide is stretched from 0% strain up to 100% strain from an initial length of 0.8 cm, at a rate of 3 mm/s. Figure 5A,B show 4 cycles of this deformation, with a hysteresis below 10% seen in Figure 5B, corresponding to an integrated hysteresis area of 2.71 μA·%. The dominant source of this hysteresis is the relaxation time of the PDMS material. Indeed, at higher strain rates the hysteresis increases significantly, see Appendix A. As seen in Figure 5C, there is a linear response of the waveguide losses versus the strain between around 20% and 100% applied strain. However, below 10% the response appears to be more quadratic, likely due to the device not being fully flat and strain-free at the supposed 0% strain stage.

The response to strain is explained by the geometric changes in the waveguide due to the Poisson effect. That is, as the waveguide is stretched along the light propagation direction, the two orthogonal dimensions will reduce proportionally by Poisson’s ratio. This could be seen as a detrimental characteristic for artificial skin since it modifies the sensor response to external strain. However, these changes are stable and can be taken into account in the sensor design. Moreover, this strain response means these sensors can be used as a simple strain sensor with sensitivity of 0.1 dB/% strain, taken as the slope of the loss with applied strain, Figure 5C. In order to evaluate the mechanical stability of these sensors a single sensor was cycled from 0% to 100% strain 100 times over a 2000 s period (Figure 5D), showing good reproducibility with low drift during cycling.

Finally, the waveguide is characterised under bending. Bending is achieved by fixing the device in the same fashion as above, and then moving the clamps towards each other. By reducing the distance between clamps, the waveguide is forced to bend. The bending radius is calculated by taking a picture of the bent device, fitting a circle to the bent portion, and recording the radius of that circle, Appendix A. During bending, a larger portion of the light has an angle θ less than the critical angle, meaning total internal reflection is not achieved, and less light is able to propagate in the waveguide. In Figure 5E the power loss with bending radius can be seen for a 400 μm thick, 0.8 cm long waveguide. This is fitted with an exponential decay curve following the model of Marcatili [53]. This model describes power of transmitted light as $P\left(r\right)\;=\;P_{0}\exp\left(-Br\right)$ with *r* the radius of curvature and *B* is a bending loss constant. Using the fitting of this equation to the data for our waveguide in Figure 5E it is possible to estimate the bending radius at which bending losses become significant. For example, a threshold of 1% excess loss due to bending could be considered, which would require limiting the bending of the device to a minimum bending radius of 1.7 mm. As the intended application of these devices is as an artificial skin, this requirement is easily met since curvatures of the human body are far larger than this; human fingers typically have a minimum radius of curvature greater than 5 mm [54].

### 4.4. Colourimetry Strain Sensor Characterisation

An alternative mechanism for the detection of applied strain to a waveguide is direct colourimetric measurement. For linear stretching, the period of the diffraction grating, Λ, increases linearly (see Appendix A). If the previous monochromatic input light source is replaced with a polychromatic source, this change in grating period can be visualised by as a changing colour at the output grating at a set angle, following Equation (Equation 2). Here, Equation (Equation 2) is reversed, as the light travels from within the waveguide to without.

Further, the angle of the incident light wavefront is 90${}^{{}^{\circ}}$ to the normal of the grating, and the 1^st^ mode is considered, m=1. Since the refractive index of the out coupling region is n=1, the angle of out-coupled light now follows $n_{c}\sin\alpha =n_{wg}+\frac{\lambda }{\Lambda }$. Since stretching increases the value of Λ, this should cause an increase in λ. To investigate this, a new waveguide was fabricated, incorporating five additional out-coupling gratings along the length of the guiding core, seen in Figure 6A, each with an area of 1 × 5 mm. A broad spectrum, white light LED was placed at the input grating of the waveguide. A CCD camera was located 60${}^{{}^{\circ}}$ above the system, used to capture images of the out-coupling gratings for applied strain. Figure 6B shows the change in colour at the out-coupling grating for strains between 0% and 135%. A clear change in colour at the output is seen in the figure, as well as good reversibility as the strain is relaxed. Figure 6C shows the change in colour through RGB analysis of these images. It can be seen that there is a clear, highly reversible shift in the grating colour with stretching.

While such colourimetry measurements present a challenge in terms of in-situ measurements, here relying on post-hoc RGB analysis of recorded images, there are several advantages that may be leveraged. The most obvious is the simple nature of the input light; the use of polychromatic light with these devices means that a wide variety of light sources may be used, or even no additional light source at all in the case of a well illuminated environment. Another advantage is the possibility of monitoring the colour change remotely, at distances defined only by the quality of camera used. Finally, it should be noted that the use of a CCD to capture the colour changes is not mandatory, and could instead be replaced with a micro-spectrophotometer to give true wavelength measurements.

## 5. Conclusions

Optical stretchable sensors have the potential to play a key role in the fabrication of artificial skin. Unlike electrical sensors, they can have an active sensing area entirely free of metallic wires, enhancing the mechanical properties of the devices. In order for these optical skin sensors to be competitive with electrical equivalents they must be properly calibrated for different mechanical stimuli, as well as for temperature. Such requirement necessitate the fabrication of multiple optical sensors in a single unit, in a simple and low cost manner. This work demonstrated such an approach, creating multiple optical sensors based on optical waveguiding using a simple, low cost fabrication method. Coupling gratings, a key element in optical waveguides, were created using a DVD-R to provide a diffraction grating mould, eliminating the need for costly silicon moulds, and allowing rapid prototyping of different designs. This approach could be further extended to gratings of different periods through the use of CDs or Blu-ray disks.

The waveguides demonstrated here show a linear loss value of 3.2 dB/cm and coupling losses of 17 dB, values consistent with other works. These waveguides are characterised in different sensing modes, including as a compression force sensor, incorporating a foam cladding layer, and as strain sensors through modulation of both light intensity and colour. These devices are further evaluated for their bending losses, and it is found that bending losses down to a radius 1.7 mm could be considered acceptable – well inline with their use as an artificial skin sensor. While the coupling losses reported are specific to the use of a laser as the incident light source, these losses remain fixed for all measurements, and do not have an impact on the additional losses induced by a compression force, strain, or bending reported here.

Currently, work looks to develop waveguide arrays based on the same fabrication process, allowing simultaneous measurement of multiple parameters, and to improve spatial resolution. It is hoped that this straight forward, low cost, and flexible fabrication approach will the growing field of optical artificial skin.

## Figures and Tables

**Figure 1 sensors-23-00114-f001:**
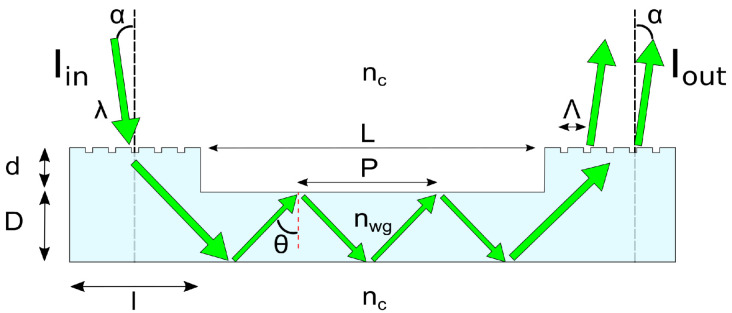
Schematic representation of the opto-geometric parameters of a planar grating coupled waveguide. Light is coupled on the left side with an integrated grating which is raised (d) from the waveguide core due to our fabrication process.

**Figure 2 sensors-23-00114-f002:**
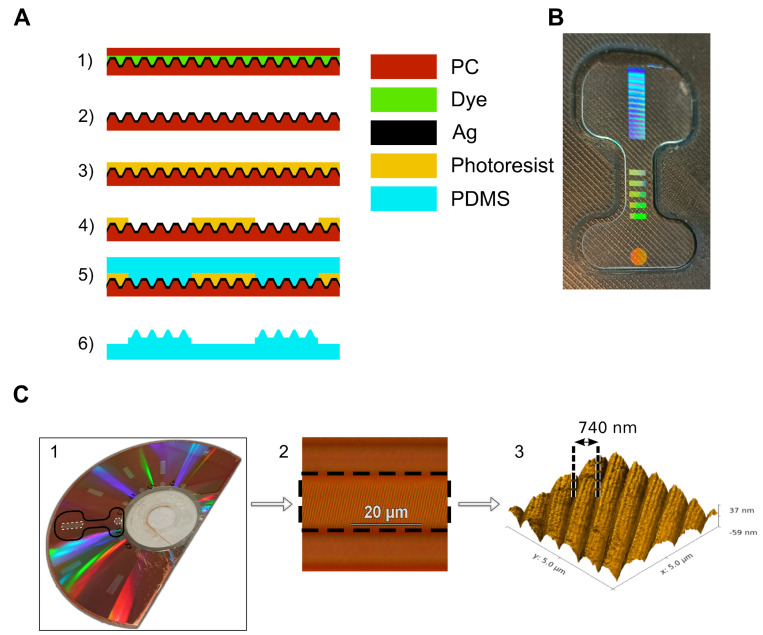
(**A**) Schematic diagram of the manufacturing process for the waveguide mould with MEMS surface micro-machining technologies; (1) DVD-R structure; (2) Separation of the top PC layer and removal of dye; (3) Spin coating of photoresist; (4) Patterning photoresist; (5) Spin coating of PDMS; (6) CO${}_{2}$ laser cutting and peel-off to reveal the final structure. (**B**) Photographic image of a stretchable planar waveguide (dog bone form) under white light illumination. (**C**) Three images of a patterned DVD-R substrate ready for waveguide fabrication. (1) Photograph of the prepared mould DVD-R with photoresist lines. The shape of a single device, to be cut by CO${}_{2}$ laser is shown in black, and the exposed grating zones are highlights with dashed white. (2) Optical microscope image of a single grating zone. The areas, top and bottom, covered with photoresist are smooth; the area in the centre, highlighted with a dashed black box, is exposed and shows the underlying DVD-R grating structure. (3) AFM image of a section of exposed DVD-R grating after removal of photoresist to define the waveguide diffraction grating. The characteristic grating distance of 740 nm is shown.

**Figure 3 sensors-23-00114-f003:**
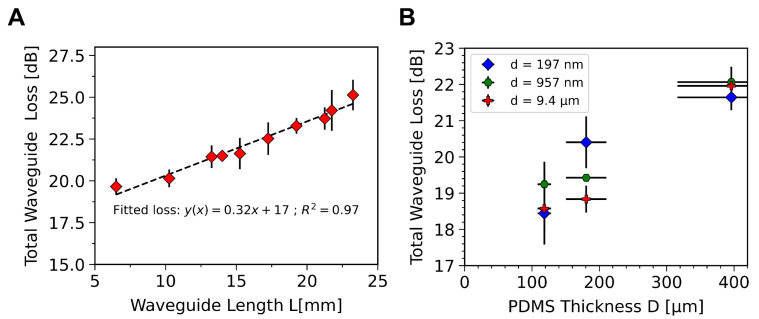
(**A**) Total loss, $A_{t}$, for 400 μm thick waveguide with varying length. (**B**) Total loss, $A_{t}$, versus the waveguide channel thickness (D) for different photoresist thicknesses (d). In both (**A**) and (**B**) the error bars show 1 standard deviation of the data from 50 measurements across 5 devices at each (**A**) waveguide length and (**B**) value of D and d.

**Figure 4 sensors-23-00114-f004:**
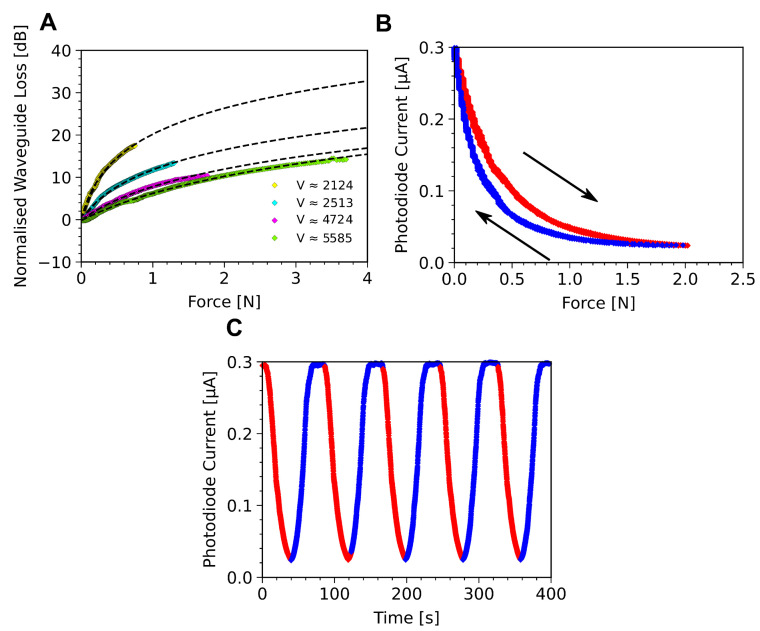
(**A**) Waveguide response for several normalised frequency values, V, calculated from Equation (Equation 1). Yellow: D = 180 μm, λ=0.532μm; Blue: D = 180 μm, λ=0.450μm; Magenta: D = 400 μm, λ=0.532μm; Green: D = 400 μm, λ=0.450μm; The black dashed lines show logarithmic fits to the data of the form $A\;=\;a\log_{b}\left(x\right)+c$, where *x* is the compression force and *a*, *b*, and *c* are freely varying fitting parameters. (**B**) Curve of the photo-current versus the compression force for a single compression force cycle, red is increasing compression force, blue is decreasing compression force. Loading and unloading achieved with a linear compression rate of 5 mm/min. The hysteresis seen has an integrated area of 37 nA·N for a single loading cycle. (**C**) Photo-current over time as compression force is applied with a 20 s pause between cycles, red and blue data correspond to part (**B**).

**Figure 5 sensors-23-00114-f005:**
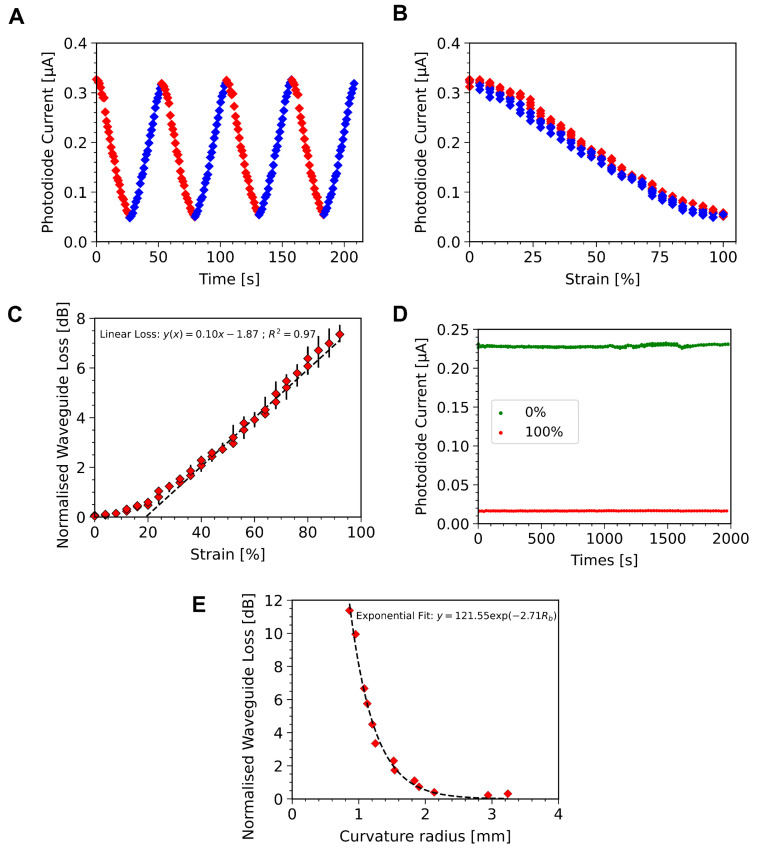
(**A**) Photo-current over time for four stretching cycles between 0% and 100% for a single device. (**B**) Photo-current against strain over four stretching cycles, stretching shown in red, relaxing in blue. Hysteresis is evaluated at around 10% for the same devices as (**A**), with an integrated current of 2.71 μA·% over a single strain cycle. (**C**) Calculated power loss in the waveguide with strain averaged over 4 cycles for the same device a (**A**), showing a linear increase in loss with strains above 20%, represented by the fitted straight line, dashed black. Error bars show the spread of data over four loading and unloading cycles of a single device. (**D**) Measured photo-current from a new device over 100 cycles of stretching from 0% to 100%, current at 0% stretch in green and 100% stretch in red. (**E**) Calculated power loss in a 400 μm thick waveguide with bending radius (red diamonds) and fitted exponential decay curve (dashed black line).

**Figure 6 sensors-23-00114-f006:**
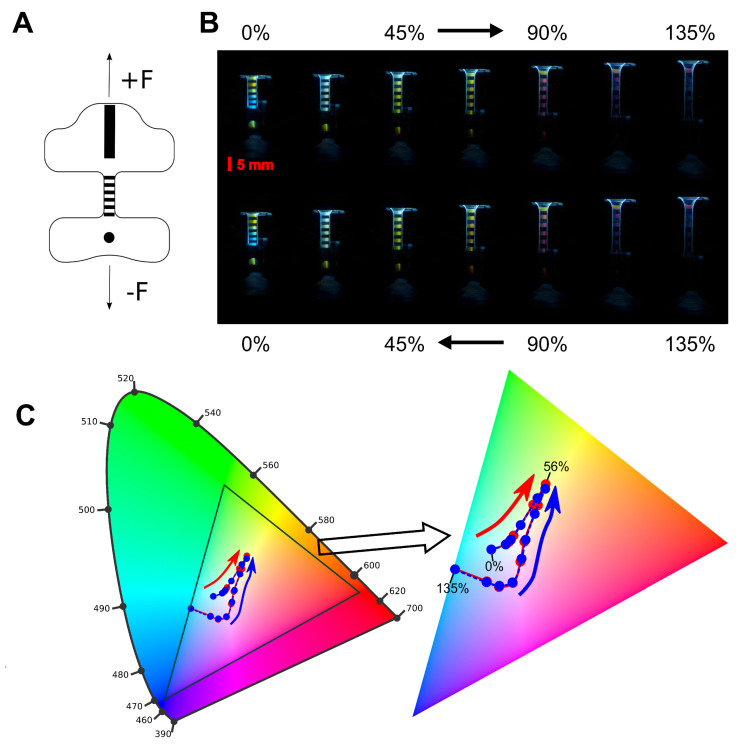
(**A**) Schematic diagram of the waveguide used for colourimetry measurements, the grating areas shown in black, a white light LED is positioned over the large grating at the top. Force is applied uniaxially. (**B**) Change in colour at the out-coupling gratings of a waveguide with applied strains between 0% and 135% using a white light input. (**C**) RGB analysis of the images in part (**B**). Extracted RGB values for each image are projected on a CIE xy chromaticity diagram, left, with an enlargement of the RGB region, right. Points are shown for stretching in red, and relaxing in blue. The position of the points while stretching and relaxing show high correlation due to the good repeatability of the method. True colour wavelengths are shown around the outside of the chromatic diagram, and the RGB colour limits are represented by the black triangle. Arrows show the direction of movement of the points with strain, red for stretching, blue for relaxing, and some strain values are labelled on the right.

## Data Availability

Data is available from the authors upon reasonable request.

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
