# Peer review of "Design, Fabrication and Characterisation of Multi-Parameter Optical Sensors Dedicated to E-Skin Applications"

_sensors, 2022, doi:10.3390/s23010114_

Round 1
Reviewer 1 Report
- If the Figure 1 is referred in text at 3rd page it is recommended, if possible, to be on the same page.
- After equation (2) is no needed to explain the parameters because they have already been presented on page 3, except ”Λ”.
- What are n1, n2 and θs in figure 1? Can be explained in text?
- The text of Figure 2 can be reduced to the main steps, and the details can be placed in text and referred as Figure 2A, figure 2B, Figure 2C.
- After Figure 2, the ”Results” can be continued and the Figure 3 can be moved to page 7.
- at 263, the surface topology parameters are referred as average roughness (Ra or Sa), root mean square roughness (Rq or Sq), maximum height of surface (Rz or Sz)
- at 250-251, how the force was applied? normal to the surface? how about the velocity? Also, Figure 5 can be moved to page 9.
Reviewer 2 Report
This paper designed and fabricated a flexible artificial skin consisting of an optical waveguide. And its characterisation has been discussed theoritically and experimentally. I think it can be accepted by Sensors after adressing the following questions.
1. Photographs of the sensor before and after stretching should be given.
2. According to the title of this paper, as a kind of artificial skin, the possible applications of the sensor need to be demonstrated.
Reviewer 3 Report
Authors proposed a stretchable polydimethylsiloxane (PDMS) elastomer waveguide with embedded gratings, replicated from DVD-Rs. The paper need a major revision but the authors are strongly encouraged to do the revision because the paper is interesting and suitable for publication in MDPI sensors.
The paper needs to be improved in terms of experimental section. Information are missing as suggested by the following:
- In the experimental section, refers to figure 2a as 2a-1, 2a-2, …, 2a-6 to clearly explain the fabrication method.
· - P.5 R.177-182: the DVD-R is a Verbatim AZO DVD. How the authors could infer on the manufacturing process of the DVD, as aluminum coating deposition by sputtering?
· - P.5 R.183-197: In this section are missing all the information about experiments. Appendix could be transferred here.
· - There are missing information about measurements in the experimental section. Measurement of losses vs. PDMS length and thickness, Measurement of losses and photodiode current vs. force, Measurement of losses and photodiode current vs. strain, Measurements of colorimetry. Clearly insert all experimental information about the compression of the sensor in the experimental section.
- Figure 2B and 2C are never mentioned in the text. Is figure 2B the dog bone form? If yes, write in the caption. Figure 2C, rename the three images with C 1, 2, and 3, and mention in the text as Figure 2C.1, 2C.2, and 2C.3. In figure 2C left is not clear where is photoresist or not. The same in figure 2C centre, the black box refers to are on which there is not photoresist? Figure 2C right, are missing all information about the path, presumably the periodic path should have a pitch of 740 nm?
- Figure 3 A-B: how many samples refer the standard deviation bars?
- P.7 R.220-224: How the diffraction grating are better coupling light with the waveguide and it is influencing the performance of the waveguide in terms of losses? A waveguide without diffraction grating should have higher or lower losses? An experiment should be performed to demonstrate the concept.
- P.7. R.228-238: authors say that the influence of d is not relevant compared to D as demonstrated by experimental results. In my opinion this could be related to the real thickness achieved for d and D. Authors should provide exacted values for d and D achieved from the manufacturing process and try to check some experimental trends. What is the process reproducibility in terms of d and D achieved thicknesses: STD and AVG over several replication of the process.
- P.7 R.237-238: “This reduction in losses for thinner waveguides is clearly associated with a reduction of guided light power.” It is not clear for me. The reduced guided light power should increase the wavelength losses if a constant input power is selected.
- P.7 R.244-267: This waveguide has no cladding, so authors say that an object in the proximity of the waveguide change the optical properties of the waveguide more than a pressure due to the change of the refractive index n2. So the authors are proposing the waveguide as pressure sensor for s-skin. How they think that a pressure could be applied in a real case without changing the optical properties of the cladding and then the refractive index n2? Authors have to focus this point!
- P.7 R.258-268: Authors proposed the use of a PE sponge to apply pressure to the sensor. Are similar experiments never did by other groups on the literature? By applying a pressure to the foam, authors are applying a pressure to the waveguide changing its geometric properties but they are also changing the equivalent refractive index of the cladding by reducing the porosity of the foam… What is the effect they see in the variation of the waveguide optical performance?
- Figure 4A: absolutely no clear what is V= 2124, …, 5278. Explain in caption what are coloured solid lines and black doshed lines.
- Figure 4B: It is preferred to refer to this curve not with the name “hysteresis curve”. The hysteresis is a property of the loading and unloading superimposed curved but not the name of the characteristic curve. Refer with the name photodiode current vs. force characteristic. Report the loading and onloading speed (mm/min). Report the area (uA N) of the photodiode current vs. force characteristic as measure of hysteresis. How authors can infer that the hysteresis is related to the PDMS waveguide or the PE foam? Check this publication for hot to describe these experiments: A. Paghi, et al., “In situ controlled and conformal coating of polydimethylsiloxane foams with silver nanoparticle networks with tunable piezo-resistive properties,” Nanoscale Horizons, vol. 7, no. 4, pp. 425–436, 2022.
- In conclusion, I think that if experiments in section 4.2 are not justified should be removed from the paper in the next resubmission before publication.
- Section 4.3: Describe well this section in experimental section for stretching and bending.
- Figure 5B: Report the area (uA) of the photodiode current vs. strain characteristic as measure of hysteresis for two consecutive loading and unloading cycles.
- Figure 5C: how many samples refer the standard deviation bars?
- Figure 5A and 5D: data of 0 and 100 % are not in agreement. Why?
- Section 4.4.: Describe manufacturing process of the device in the experimental section. Describe experiments. In Figure 6B it seems that not all the diffraction gratings has the same colours (see 0% strain); why? So, if they did not have the same colour, where the colour of the image is analyzed?
- P.2 R55-67: Referring to introduction, authors refers to “island architecture”. I never road this definition. Usually when we refer to nanocomposite materials we refers to hosting materials which embedded hosted materials on the surface or in the volume. So I agree with the authors for the second part of this section but they should review the first part of the section. Here some papers from which they can read some interesting information to improve this section:
A. Paghi, et al., “In situ controlled and conformal coating of polydimethylsiloxane foams with silver nanoparticle networks with tunable piezo-resistive properties,” Nanoscale Horizons, vol. 7, no. 4, pp. 425–436, 2022.
S. Mariani et al., “4D Printing of Plasmon‐Encoded Tunable Polydimethylsiloxane Lenses for On‐Field Microscopy of Microbes,” Adv. Opt. Mater., vol. 10, no. 3, p. 2101610, Feb. 2022.
J. Pinto, et al., Antibacterial Melamine Foams Decorated with in Situ Synthesized Silver Nanoparticles, ACS Appl. Mater. Interfaces, 2018, 10(18), 16095–16104.
A. M. Khalil, et al., Gold-decorated polymeric monoliths: Insitu vs ex-situ immobilization strategies and flow through catalytic applications towards nitrophenols reduction, Polymer, 2015, 77, 218–226.
Round 2
Reviewer 3 Report
Authors implemented the large part of comments I did. Unfortunatly, answers to some of my questions are not satisfactionary referring to the role of the PE sponge on the optomechanical perfomance of the waveguide. I think that the use of a PE sponge to apply a force to a waveguide could be an interesting method to fine tune the applied stress value, but it is not possible extrapolate the optomechanical performance of the waveguide from the optomechanical permorfomance of the waveguide-PE sponge system.So, I think that the paper sounds good also without the section 4.2 and this section shoud be removed from the manuscript.
After the removal, the paper is suitable for publication.
